# Controllable Music Loops Generation with MIDI and Text via Multi-Stage Cross Attention and Instrument-Aware Reinforcement Learning

## ABSTRACT

The burgeoning field of text-to-music generation models has shown great promise in their ability to generate high-quality music aligned with users' textual descriptions. These models effectively capture abstract/global musical features such as style and mood. However, they often inadequately produce the precise rendering of critical music loop attributes, including melody, rhythms, and instrumentation, which are essential for modern music loop production. To overcome this limitation, this paper proposed a Loops Transformer and a Multi-Stage Cross Attention mechanism that enable a cohesive integration of textual and MIDI input specifications. Additionally, a novel Instrument-Aware Reinforcement Learning technique was introduced to ensure the correct adoption of instrumentation. We demonstrated that the proposed model can generate music loops that simultaneously satisfy the conditions specified by both natural language texts and MIDI input, ensuring coherence between the two modalities. We also showed that our model outperformed the state-of-the-art baseline model, MusicGen, in both objective metrics (by lowering the FAD score by 1.3, indicating superior quality with lower scores, and by improving the Normalized Dynamic Time Warping Distance with given melodies by 12%) and subjective metrics (by +2.56% in OVL, +5.42% in REL, and +7.74% in Loop Consistency). These improvements highlight our model's capability to produce musically coherent loops that satisfy the complex requirements of contemporary music production, representing a notable advancement in the field. Generated music loop samples can be explored at: https://loopstransformer.netlify.app/.

## CCS CONCEPTS

• **Computing methodologies** → **Neural networks**; *Learning latent representations*; **Artificial intelligence**.

## KEYWORDS

Text-to-Music Generation, Controllable Music Generation, Residual Vector Quantization, Loop Generation, Reinforcement Learning, Transformer

## 1 INTRODUCTION

Loops play a critical role in modern music production, enabling creators to construct complex and layered compositions by repeatedly using sound loops. These loops can include melodies, rhythms, or textures, and are particularly popular among DJs, electronic music producers, and contemporary musicians. In the process of music creation, loops often act as the foundation for musical accompaniment, with the final musical piece enriched through the layering of various elements. Essential to this process is the precise manipulation of melody, rhythm, BPM, key, and instrumentation within the loops.

This study explores the generation of musical loops from text and MIDI inputs, which is particularly relevant during the music creation process. Creators commonly employ Digital Audio Workstations (DAWs) to translate their creative requirements—such as melody, rhythm, BPM, key, and velocity—into MIDI format. By integrating textual prompts with specific MIDI settings, it becomes possible to produce tailored loops. This approach not only accelerates the experimental and exploratory phases of music creation but also enhances its interactivity, offering composers a richer and more malleable creative landscape.

The current text-to-music generation models, such as those based on Diffusion Models [26, 29], including Noise2Music [12], AudioLDM 2 [19], JEN-1 [18], and MusicLDM [2]; as well as text-to-music models based on Vector Quantization (VQ) [31] and Residual Vector Quantization (RVQ) [4, 38], such as Jukebox [5], MusicLM [1], and MusicGen [3], can generate music that aligns with the user's textual input. While these models effectively capture abstract/global musical features like style and mood, they often inadequately address the precise rendering of critical music loop attributes, including melody, rhythms, and instrumentation, which are essential for modern music loops production.

To overcome the aforementioned challenges, we introduce Loops Transformer (model architecture shown in Figure 1), a controllable music loops generation model capable of generating high-quality loops given both textual descriptions and MIDI inputs. To better integrate text and MIDI, we propose a novel Multi-Stage Cross Attention mechanism that combines text embeddings (from the T5 model [24] and our Text-to-MIDI Transformer) and MIDI embeddings (from our MIDI Transformer).

The model's training objective is to generate high-quality 32 kHz music loops by modeling multiple parallel streams of discrete audio tokens, which are obtained through the Encodec [4] using Residual Vector Quantization (RVQ). To enhance the quality and usability of the generated loops (making them more suitable for the characteristics of loopable music), we improve upon the Codebook Interleaving Pattern [1, 3] (illustrated in Figure 2). Based on the Delay Pattern [3], we introduce $s$ and $e$ tokens in the Codebook

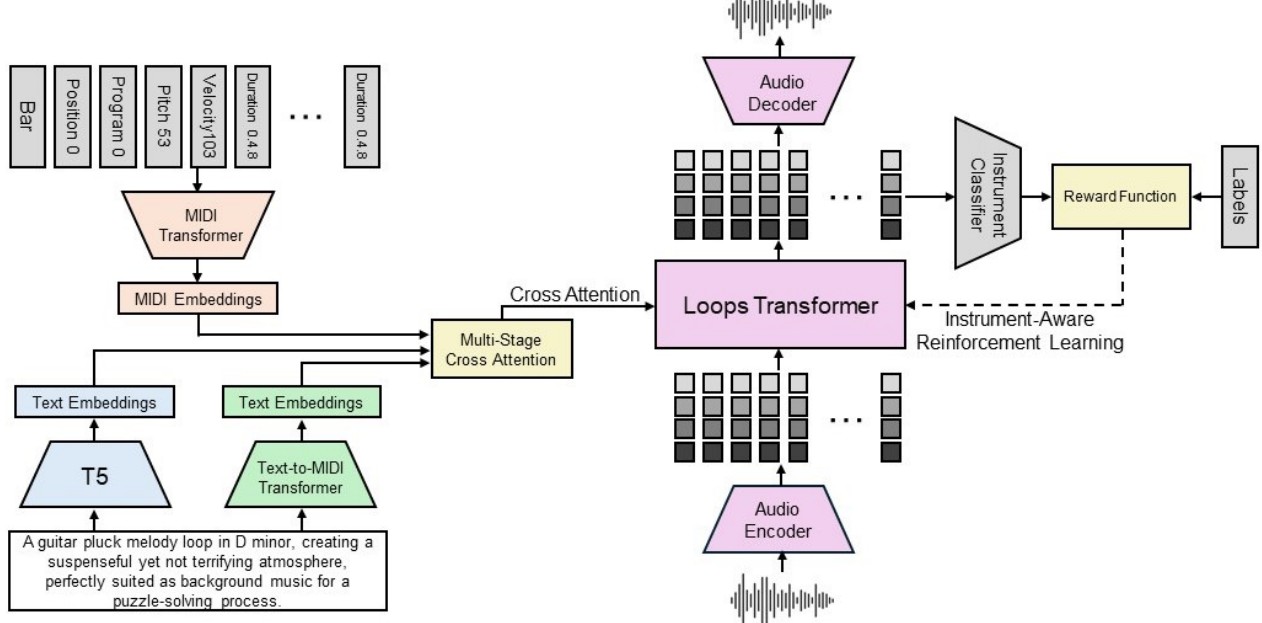

**Figure 1: The Overview of The Loops Transformer Architecture for Training.**

Interleaving Pattern to enable the model to explicitly learn the start and end of a loop segment and generate results that can be seamlessly looped.

Furthermore, during the pre-training stage, we introduce a Loop Shift mechanism for data augmentation, i.e., sampling the $s_1$ and $s_2$ as two randomly sampled integer values that determine the number of time steps to be added before the start token $s$ and after the end token $e$, respectively. The Loop Shift data augmentation mechanism is designed based on the characteristics of loops, as an ideal loop should have content generated after the start token $s$ that can form a continuous playback loop with the content before the end token $e$.

Moreover, to improve the model's instrumentation capabilities and performance, i.e., ensuring that the instruments used in the generated loops match the textual prompts, we draw inspiration from the Reinforcement Learning from Human Feedback (RLHF) [22, 30] approach that has flourished in the development of large language models (LLMs). We propose a novel Instrument-Aware Reinforcement Learning training strategy, introducing a scoring model for instrumentation evaluation. We feed the loops generated by the Loops Transformer into this scoring model for multi-label instrument classification to obtain the reward to update the Loops Transformer's model parameters.

We conduct the extensive evaluation, and the experimental results demonstrate that, compared to the state-of-the-art baseline model (MusicGen [3]), our approach significantly improves melody matching by 12% in Normalized DTW Distance and shows notable gains in both objective and subjective evaluations. Notably, in the human evaluation conducted by evaluators with experience in using or creating loops, our model achieves significant improvements in subjective ratings, such as overall quality (OVL): +2.56%, relevance to the text input (REL): +5.42%, and Loop Consistency: +7.74%. This demonstrates our model's ability to produce musically consistent loops that meet the detailed requirements of contemporary music production.

To summarize our contributions:

**(1)** We introduce Loops Transformer, a novel controllable music loops generation model that integrates text and MIDI inputs using a Multi-Stage Cross Attention mechanism and an improved Codebook Interleaving Pattern with Loop Shift data augmentation for seamless and musically consistent loops generation.

**(2)** We propose an Instrument-Aware Reinforcement Learning training strategy to enhance the model's ability to generate loops with instruments matching the textual prompts.

**(3)** We demonstrate the effectiveness of our proposed methods through extensive experiments, significantly outperforming state-of-the-art baselines in generating high-quality, relevant, and consistent music loops, as validated by objective and subjective metrics. Furthermore, we conduct comprehensive ablation studies to analyze the impact of each key component and their individual contributions to the overall performance of our model.

## 2 RELATED WORK

In this section, we review the relevant literature in the key areas of loop music generation. We discuss the various approaches and studies in relation to our proposed Loops Transformer model.

### 2.1 MIDI Representation

The Transformer based models [32] has inspired various tokenization methods for effectively modeling the sequences of MIDI data.

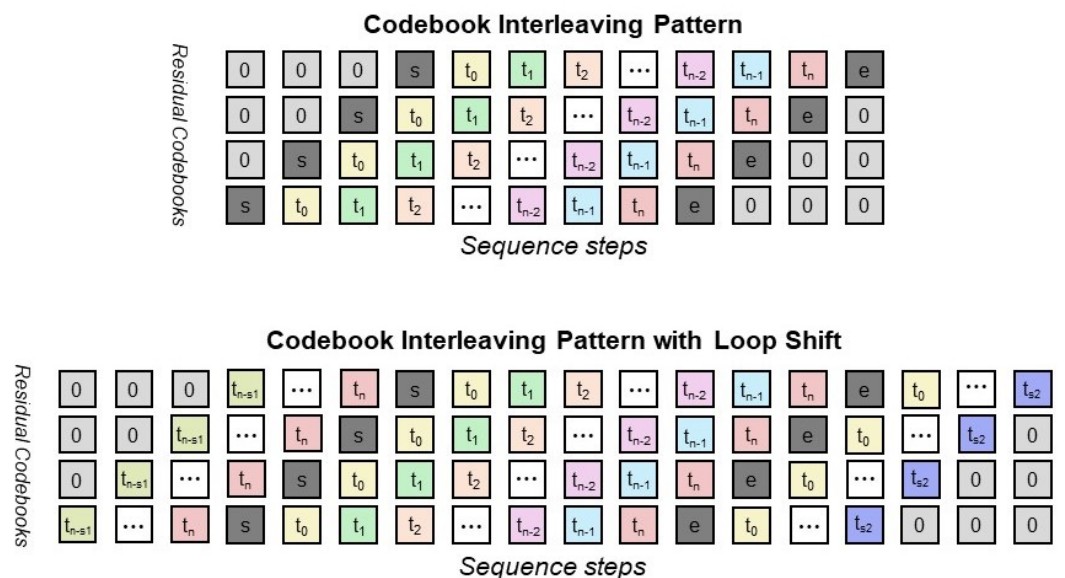

**Figure 2: The Codebook Interleaving Pattern of Loops Transformer.**

REMI [14] encodes notes as a combination of Pitch, Velocity, Duration, Bar, and Position tokens, enabling Transformers to better understand music structure. MIDI-Like [21] directly converts MIDI messages into tokens, a method employed by models like Music Transformer [11] and MT3 [9]. CPWord [10] builds upon REMI, reducing sequence length by combining embeddings.

Our work focuses on diverse MIDI data, including single-track and multi-track compositions with various instruments. We leverage REMI+ [33], an extension of REMI designed for general multi-track, multi-signature symbolic music sequences, introducing Program tokens for efficient representation of multiple instruments. We employ REMI+ as the tokenizer of our models.

## 2.2 Loop Generation

There are a few studies that have explored the topic of loop generation. [37] proposes an unconditional loop generation model based on StyleGAN2 [15] to create 2-second loops, while [39] presents a system that integrates ChatGPT with music generation models [3, 8] to enable interactive and iterative music creation through a multi-round dialogue interface.

However, these works do not fully address the fundamental aspects of loops, which are essential building blocks in modern music production. Loops allow creators to construct intricate compositions by repeatedly using melodic, rhythmic, or textural elements, with precise control over melody, rhythm, and instrumentation being crucial to the creative process.

Our work focuses on generating musical loops from text and MIDI inputs. By combining textual prompts with specific MIDI settings, our approach facilitates the creation of customized loops, expediting the experimentation and exploration phases of music production while providing composers with a more interactive and adaptable creative environment.

## 2.3 Text to Music Generation

Text-to-music generation models have made significant progress in recent years, with approaches based on Diffusion Models [26, 29], such as Noise2Music [12], AudioLDM 2 [19], JEN-1 [18], and MusicLDM [2], as well as those utilizing Vector Quantization (VQ) [31] and Residual Vector Quantization (RVQ) [4, 38], like Jukebox [5], MusicLM [1], and MusicGen [3]. While these models excel at capturing global musical features such as style and mood, they often struggle to precisely render critical attributes for music loop generation, including melody, rhythm, and instrumentation.

Recent works have aimed to enhance the controllability of generated music by incorporating additional features. [3] explores melody conditioning using chromagrams, while [34] proposes Music ControlNet, a diffusion-based model with time-varying controls over melody, dynamics, and rhythm.

Unlike previous works [3, 34] that rely on chromagrams for melody conditioning, our approach harnesses MIDI as a more comprehensive conditional input, encompassing pitch, velocity, duration, bar, position, multi-tracks, and multi-instruments information. We introduce Loops Transformer, a controllable music loops generation model that integrates text and MIDI using a novel Multi-Stage Cross Attention mechanism, improves upon the Codebook Interleaving Pattern [1, 3] for seamless looping, and employs a Loop Shift data augmentation mechanism and an Instrument-Aware Reinforcement Learning strategy to generate high-quality, relevant, and consistent music loops that match textual prompts and MIDI conditions. Extensive evaluations demonstrate our model's superiority compared to state-of-the-art baselines.

## 3 METHODOLOGY

The overview of the Loops Transformer architecture is shown in Figure 1. The following subsections will provide explanations for each component.

### 3.1 Textual Representation and Embeddings

In the proposed architecture, the conversion of text into an embedding representation marks the commencement of the process. We denote a text sequence by $s$, which is input into a transformer-based encoder, specifically $\mathcal{T}_{\text{text}}$. This encoder maps the input text into a high-dimensional vector space, yielding the primary text embeddings $\mathbf{T}_s = \mathcal{T}_{\text{text}}(s)$.

In this context, $\mathbf{T}_s$ materializes as a sequence of vectors in $\mathbb{R}^{l \times d_t}$, where $l$ characterizes the sequence length and $d_t$ defines the dimensionality of the embedding space. The encoding procedure adheres to the approach by [3], utilizing the T5 [24] encoder as $\mathcal{T}_{\text{text}}$.

Advancing to integrate text with MIDI data, we introduce a secondary encoding step using the Text-to-MIDI Transformer encoder $\mathcal{T}_{\text{t2m}}$, which has been pre-trained on our MIDI Loops Dataset (with text and MIDI paired data). This step is designed to refine the text embeddings by incorporating musical context. Thus, the secondary text embeddings, denoted as $\mathbf{T}_{s'} = \mathcal{T}_{\text{t2m}}(s)$, are generated.

The secondary embeddings $\mathbf{T}_{s'}$ not only retain the encapsulated textual information but also incorporate a musical context, resulting in a sequence of vectors in $\mathbb{R}^{l \times d_{t'}}$. The dimension $d_{t'}$ is representative of the embedding space.

In this work, the Text-to-MIDI Transformer leverages the same encoder architecture and tokenizer as utilized by the T5-base model [24]. We employ REMI+ for MIDI tokenization, as detailed in [33]. To address the inherently longer sequence lengths of MIDI tokens relative to text, we adopt the Light Decoder architecture described by [36], modifying the T5 decoder to accommodate these differences. This approach reduces the computational and memory consumption, thus enhancing our model's capability to process extended sequences.

### 3.2 MIDI Representation and Embeddings

Concurrently, the MIDI input $m$, consisting of the sequence of tokens with their corresponding attributes such as pitch, velocity, and duration, are processed by a MIDI Transformer $\mathcal{T}_{\text{MIDI}}$ with the REMI+ [33] tokenizer. This results in a series of MIDI embeddings $\mathbf{M}$, which are represented as $\mathbf{M} = \mathcal{T}_{\text{MIDI}}(m)$.

with each $\mathbf{M} \in \mathbb{R}^{l_m \times d_m}$, where $l_m$ is the length of MIDI representation sequence and $d_m$ is the MIDI embedding size.

### 3.3 Multi-Stage Cross-Attention Mechanism

The foundational principle of the proposed Multi-Stage Cross-Attention Mechanism involves leveraging the embeddings generated by the Text-to-MIDI Transformer as intermediaries. This mechanism encompasses two distinct stages of cross-attention[32] operations followed by a final integration stage:

**Textual Embeddings Cross-Attention:** In the initial stage, cross-attention is applied between the primary text embeddings $\mathbf{T}_s$ and the secondary text embeddings $\mathbf{T}_{s'}$, resulting in a new sequence of embeddings, denoted by $\mathbf{C}_{\text{txt}}$, which integrates the textual content with the encoded musical context as:

$$\mathbf{C}_{\text{txt}} = \text{CrossAttention}(\mathbf{T}_s, \mathbf{T}_{s'}).$$

**MIDI Embeddings Cross-Attention:** Subsequently, a cross-attention operation is performed between the MIDI embeddings $\mathbf{M}$ and the secondary text embeddings $\mathbf{T}_{s'}$. This generates another new sequence of embeddings, $\mathbf{C}_{\text{MIDI}}$, that merges the musical information with the contextual data from the text as:

$$\mathbf{C}_{\text{MIDI}} = \text{CrossAttention}(\mathbf{M}, \mathbf{T}_{s'}).$$

**Final Cross-Attention Stage:** The final stage involves a cross-attention operation between the newly generated embeddings sequences $\mathbf{C}_{\text{txt}}$ and $\mathbf{C}_{\text{MIDI}}$, producing the ultimate embedding sequence $\mathbf{C}_{\text{final}}$ that encapsulates both textual and musical contexts comprehensively as:

$$\mathbf{C}_{\text{final}} = \text{CrossAttention}(\mathbf{C}_{\text{txt}}, \mathbf{C}_{\text{MIDI}}).$$

The cross-attention operation is mathematically defined as:

$$\text{CrossAttention}(\mathbf{Q}, \mathbf{K}, \mathbf{V}) = \text{Softmax}\left(\frac{\mathbf{Q}\mathbf{K}^T}{\sqrt{d_k}}\right)\mathbf{V}$$

where $\mathbf{Q}$ represents the query matrix from one set of embeddings, $\mathbf{K}$ and $\mathbf{V}$ represent the key and value matrices from another set of embeddings, and $d_k$ is the dimensionality of the key vectors, which is used for scaling.

The Multi-Stage Cross-Attention Mechanism aims to seamlessly fuse textual and musical data through successive layers.

### 3.4 Loops Transformer

The Loops Transformer serves as the core generative model of our architecture, leveraging the rich contextual embeddings $\mathbf{C}_{\text{final}}$ to guide the autoregressive generation of discrete audio tokens. Following the approach of the MusicGen Transformer introduced by [3], we employ the 32 kHz EnCodec [4] to convert audio into a discrete representation.

The primary objective of the Loops Transformer is to model the conditional probability distribution over a sequence of discrete audio tokens $\mathbf{A} = (a_1, a_2, \ldots, a_M)$, conditioned on the fused textual and musical representation $\mathbf{C}_{\text{final}}$:

$$p(\mathbf{A}|\mathbf{C}_{\text{final}}) = \prod_{j=1}^{M} p(a_j|a_{<j}, \mathbf{C}_{\text{final}}) \tag{1}$$

where $a_j$ represents the audio token at index $j$, and $a_{<j}$ denotes the sequence of all preceding audio tokens. The Loops Transformer employs a transformer decoder architecture, tasked with predicting the subsequent audio token based on the previously generated tokens and the conditioning information embedded within $\mathbf{C}_{\text{final}}$.

During the training phase, the objective is to maximize the log-likelihood of the audio token sequence, conditioned on the fused representation:

$$\mathcal{L} = \sum_{j=1}^{M} \log p(a_j|a_{<j}, \mathbf{C}_{\text{final}}) \tag{2}$$

The Codebook Interleaving Pattern of Loops Transformer (shown in Figure 2) is a key component of the model, employed during both pre-training and fine-tuning stages. In the pre-training phase, the Codebook Interleaving Pattern with Loop Shift is used for data augmentation, where randomly sampled time steps ($s_1$ and $s_2$) are added before the start token $s$ and after the end token $e$. This Loop

Shift mechanism ensures that the content generated after the start token $s$ can seamlessly loop back to the content before the end token $e$, mimicking the characteristics of an ideal loop. During fine-tuning, the standard Codebook Interleaving Pattern is applied, incorporating the start token $s$ and end token $e$ for autoregressive modeling, which maintains the sequence's integrity while making predictions.

To reconstruct the final generated musical audio, the discrete audio tokens produced by the Loops Transformer are decoded back into a continuous waveform representation using the decoder component of the EnCodec model. By leveraging the high-fidelity audio compression capabilities of EnCodec and the expressive power of the transformer architecture, the Loops Transformer enables the generation of high-quality and controllable music that aligns with the given textual and musical conditions.

It is worth noting that, in order to enable the Loops Transformer to generate music not only based on the integrated text and MIDI conditions but also solely on the text description, we introduce a training strategy that randomly replaces $\mathbf{C}_{\text{final}}$ with $\mathbf{C}_{\text{txt}}$ during the training process. Specifically, with a probability of 1/3 (as set in our experiments), we substitute $\mathbf{C}_{\text{final}}$ with $\mathbf{C}_{\text{txt}}$, allowing the model to learn how to generate music based only on the text condition, i.e., in the absence of MIDI information. This training strategy endows the Loops Transformer with greater flexibility,

## 3.5 Instrument-Aware Reinforcement Learning

To further enhance the Loops Transformer's ability to generate music loops that align with the specified instruments in the textual prompts, we propose an Instrument-Aware Reinforcement Learning approach. This method leverages the Proximal Policy Optimization (PPO) algorithm [28] and a novel reward function based on an instrument classification model.

Let $\pi_\theta$ denote the policy of the Loops Transformer parameterized by $\theta$, and let $R(\mathbf{A})$ represent the reward function for a generated discrete audio sequence $\mathbf{A}$. The objective of the reinforcement learning process is to maximize the expected reward:

$$J(\theta) = \mathbb{E}_{\mathbf{A} \sim \pi_\theta} [R(\mathbf{A})] \tag{3}$$

To compute the reward, we introduce an instrument classification model $\Phi_\psi$ parameterized by $\psi$, which is trained to predict the presence of various instruments in a given audio sequence. The model is trained using the same architecture as the Loops Transformer but with a multi-label classification objective. Let $\mathbf{y}$ denote the ground truth instrument labels for a discrete audio sequence, and let $\hat{\mathbf{y}} = \Phi_\psi(\mathbf{A})$ represent the predicted instrument probabilities. We define the reward function as :

$$R(\mathbf{A}) = \beta \cdot \exp\left(\frac{1}{N} \sum_{i=1}^{N} [y_i \log(\hat{y}_i) + (1 - y_i) \log(1 - \hat{y}_i)]\right) \tag{4}$$

where $N$ is the number of instrument classes, $y_i$ is the ground truth label for the presence of instrument $i$, $\hat{y}_i$ is the predicted probability of instrument $i$ being present, and $\beta$ is a scaling factor. This reward function is based on the binary cross-entropy loss, which measures the discrepancy between the predicted probability distribution and the true label distribution. This reward function

encourages the model to generate music loops with instrumentation that closely matches the given conditions.

To optimize the policy using PPO, we define the advantage function $\hat{A}_t$ at time step $t$ as:

$$\hat{A}_t = \sum_{k=0}^{K-1} \gamma^k R(\mathbf{A}_{t+k}) - V_\phi(\mathbf{s}_t) \tag{5}$$

where $\gamma$ is the discount factor, $K$ is the number of steps in the rollout, and $V_\phi$ is a value function parameterized by $\phi$, which estimates the expected cumulative reward from state $\mathbf{s}_t$.

The PPO objective function, $\mathcal{L}^{PPO}(\theta)$, is constructed from two primary components: the probability ratio, $r_t(\theta)$, and the advantage estimate, $\hat{A}_t$. Specifically, $r_t(\theta)$ quantifies the ratio of the new policy's probability of taking action $\mathbf{a}_t$ given state $\mathbf{s}_t$ to that of the old policy, mathematically represented as:

$$r_t(\theta) = \frac{\pi_\theta(\mathbf{a}_t|\mathbf{s}_t)}{\pi_{\theta_{\text{old}}}(\mathbf{a}_t|\mathbf{s}_t)} \tag{6}$$

With these definitions, the PPO objective simplifies to:

$$\mathcal{L}^{PPO}(\theta) = \hat{\mathbb{E}}_t \left[ \min\left( r_t(\theta)\hat{A}_t, \text{clip}\left(r_t(\theta), 1 - \epsilon, 1 + \epsilon\right) \hat{A}_t \right) \right] \tag{7}$$

where the clip function limits $r_t(\theta)$ to the range $[1 - \epsilon, 1 + \epsilon]$, with $\epsilon$ being a hyperparameter that controls the extent of clipping. This ensures modest updates to the policy, avoiding drastic deviations from the previous policy. The expectation $\hat{\mathbb{E}}_t$ averages over a batch of samples, calculating the expected value of the minimum between the unclipped and clipped objectives.

Finally, we integrate the reinforcement learning objective with the original Loops Transformer training objective to obtain the final objective function:

$$\mathcal{L}_{\text{final}} = \mathcal{L} + \beta \mathcal{L}^{PPO}(\theta) \tag{8}$$

where $\beta$ is a hyperparameter that balances the importance of the reinforcement learning objective.

By optimizing this combined objective, the Loops Transformer learns to generate music loops that not only adhere to the provided textual and musical conditions but also incorporate the desired instruments as specified in the prompts.

## 4 EXPERIMENTAL SETUP

### 4.1 Model and Hyperparameter Settings

For the text encoder $\mathcal{T}_{\text{text}}$, we employ the pretrained T5-base model [24] from the implementation[1]. The Text-to-MIDI Transformer encoder $\mathcal{T}_{\text{t2m}}$ follows the same architecture as the T5-base model but is pre-trained on our MIDI Loops Dataset (with text and MIDI paired data). We modify the T5 decoder to accommodate the longer sequence lengths of MIDI inputs, adopting the light decoder architecture as ByT5-base [36]. The $\mathcal{T}_{\text{t2m}}$ is trained for 1,000,000 steps with a batch size of 128 and a learning rate of 1e-4 using the AdamW optimizer [20] ($\beta_1 = 0.9$, $\beta_2 = 0.95$, and a weight decay of 0.1).

The MIDI Transformer $\mathcal{T}_{\text{MIDI}}$ adopts MidiTok [7][2] to implement the REMI+ Tokenization. It also follows the ByT5-base light decoder

---

[1]https://github.com/facebookresearch/audiocraft
[2]https://miditok.readthedocs.io/en/v3.0.2/

architecture [36] and is pretrained with our MIDI Loops Dataset (using MIDI data only) via an autoregressive manner, trained for 1,000,000 steps with a batch size of 128 and a learning rate of 1e-4 using the AdamW optimizer ($\beta_1 = 0.9$, $\beta_2 = 0.95$, and a weight decay of 0.1).

For the Loops Transformer, we follow the transformer decoder architecture of MusicGen-melody (1.5B)[3], as presented in [3], and utilize its pretrained weights. The Loops Transformer is first pretrained on our Audio Loops Dataset using the Codebook Interleaving Pattern with Loop Shift for data augmentation, with 1,000,000 training steps using the AdamW optimizer ($\beta_1 = 0.9$, $\beta_2 = 0.95$, and a weight decay of 0.1) and a batch size of 128, employing a cosine learning rate schedule with a warmup of 2,500 steps. The fine-tuning process spans 500,000 steps with a batch size of 128 using the AdamW optimizer ($\beta_1 = 0.9$, $\beta_2 = 0.95$, and a weight decay of 0.1). During fine-tuning, the standard Codebook Interleaving Pattern is applied. To enable the Loops Transformer to generate music based solely on text descriptions, we incorporate a training strategy that randomly replaces $C_{final}$ with $C_{txt}$ with a probability of 1/3. During the sampling process, we apply top-k sampling[6], retaining the top 250 tokens at a temperature of 1.0, as suggested by [3].

In the Instrument-Aware Reinforcement Learning setup, the instrument classification model $\Phi_\psi$ follows the same architecture as the Loops Transformer. The model is trained using the binary cross-entropy loss with a learning rate of 1e-4 and a batch size of 64 for 200,000 steps, utilizing our Audio Loops Dataset for training. For the PPO algorithm, we set the discount factor $\gamma$ to 0.99 and the clipping parameter $\epsilon$ to 0.2. The coefficient $\beta$ for balancing the reinforcement learning objective is set to 0.1. We train the Loops Transformer with the PPO objective for an additional 350,000 steps.

All models are implemented using the PyTorch library [23] and trained on 8 NVIDIA A100 GPUs with 80GB memory each.

## 4.2 Datasets

We train, validate, and test our model using licensed music loops (including MIDI and wav files) created by professional musicians. These loops are accompanied by relevant musical information (e.g., BPM, key, instruments). We generate corresponding textual descriptions using ChatGPT[4] and Claude[5], taking into account the descriptions of loop sample packs or loop bundles and the provided musical information. Human annotators verify and modify each description with its corresponding loop as necessary to ensure quality. The following is an example of a textual prompt of a loop: "*A heartfelt piano loop in A minor at 140 BPM, weaving a narrative of delicate emotions and tender moments, ideal for storytelling and introspective journeys.*".

We organize these loop data into three datasets for different stages of use. The **MIDI Loops Dataset**, consisting of 16,432 MIDI format loops (including single-track and multi-track data), is used to pre-train our MIDI Transformer and Text-to-MIDI Transformer. The **Audio Loops Dataset**, comprising 18,182 wav format loops (95.9 hours), is used for pre-training the Loops Transformer model.

The **MIDI-Audio Paired Loops Dataset** contains 10,018 loops (Train: 8,129, Validation: 932, Test: 957), with MIDI and wav pair data for each loop, along with corresponding textual descriptions. We use this dataset to fine-tune the proposed model. To ensure the correctness and evaluability of the experiments, we carefully confirm that the test dataset comes from different sources (i.e., different packs, bundles, and producers) than the training and validation data of these three datasets (i.e. the MIDI, the Audio, and the MIDI-Audio Paired Loops Dataset), ensuring no artist overlap between the test set and the training and validation sets. This guarantees that the test data and the data used for pre-training and fine-tuning are from completely different sources, without any overlap.

We use the test set of the MIDI-Audio Paired Loops Dataset for objective metric evaluation and randomly sample 150 test samples from the test set for subjective metric and the Instrumentation Score (IS) evaluation.

## 4.3 Evaluation

**Baseline Models.** In this work, we primarily compare the proposed model with MusicGen [3][6], a state-of-the-art text-to-music model capable of generating music conditioned on both text and melody. MusicGen serves as a strong baseline for our evaluation, as it represents the current best-performing model in the field of text-to-music generation. Since MusicGen can leverage the given audio for melody conditioning, we convert our MIDI input to audio using FluidSynth[7] to provide the melody condition for MusicGen.

**Objective Evaluation Metrics.** We evaluate the performance of the generated audio using three objective metrics: Fréchet Audio Distance (FAD) [16], CLAP score [13, 35], and Normalized Dynamic Time Warping Distance (Norm. DTW Distance) [27].

FAD measures the distance between the distributions of the generated audio and real audio in the feature space of a pre-trained VGGish model. We compute FAD using the official TensorFlow implementation[8]. A lower FAD score indicates that the generated audio is closer to real audio in terms of its distribution, suggesting better perceptual quality.

The CLAP score, calculated using the official pre-trained CLAP model[9], measures the semantic alignment between the generated audio and the corresponding text description. A higher CLAP score suggests that the generated audio effectively captures the semantic content of the input text.

To quantify the similarity between the given MIDI melody and the generated audio, we introduce the Norm. DTW Distance. First, we convert the ground truth MIDI to audio using FluidSynth, employing the same process as we did for generating the melody condition for MusicGen. We then convert the ground truth audio and the generated audio into chromagrams. We apply an argmax operation to the chromagram, preserving only the most prominent pitch class in each frame, creating a frame-wise one-hot representation of the 12 semitones. We then compute the Normalized Dynamic Time Warping Distance between the two chromagram sequences. For each sample, we calculate the Norm. DTW Distance using a 10-second segment of the ground truth and generated audio,

---

[3]https://github.com/facebookresearch/audiocraft/blob/main/docs/MUSICGEN.md
[4]https://chat.openai.com/
[5]https://claude.ai/

[6]https://huggingface.co/facebook/musicgen-melody
[7]https://www.fluidsynth.org/
[8]https://github.com/google-research/google-research/tree/master/frechet_audio_distance
[9]https://github.com/LAION-AI/CLAP

**Table 1: Evaluation results demonstrating the performance of the Loops Transformer in comparison to MusicGen, a leading state-of-the-art baseline model, across various configurations. The evaluation is based on objective metrics such as Fréchet Audio Distance (FAD$_{vgg}$), CLAP score, and Normalized Dynamic Time Warping Distance (Norm. DTW Distance), with optimal performance indicated by lower FAD and Norm. DTW Distance, and higher CLAP scores. Subjective assessments are made through metrics including Overall Quality (OVL.), Relevance to Text Input (REL.), and Loop Consistency, on a scale of 1 to 100, where higher scores signify superior outcomes.**

| Model | FAD$_{vgg}$ ↓ | CLAP ↑ | Norm. DTW Distance ↓ | OVL. ↑ | REL. ↑ | Loop Consistency ↑ |
|---|---|---|---|---|---|---|
| MusicGen (text) | 4.1 | **0.30** | - | 84.44±1.17 | 81.12±1.42 | 77.41±1.62 |
| MusicGen (text + melody) | 3.8 | 0.27 | 0.23 | 85.69±1.09 | 80.54±1.33 | 80.23±1.51 |
| MusicGen (text + half melody) | 3.8 | 0.27 | 0.31 | 85.66±1.11 | 80.32±1.36 | 81.05±1.45 |
| Loops Transformer (text) | 2.8 | 0.22 | - | 86.32±1.15 | 84.28±1.27 | 86.98±1.29 |
| Loops Transformer (text + midi) | 2.6 | 0.22 | **0.11** | **88.25**±0.96 | **85.96**±1.08 | **88.79**±1.11 |
| Loops Transformer (text + half midi) | **2.5** | 0.22 | 0.13 | 87.79±1.02 | 85.66±1.19 | 88.21±1.23 |

normalizing the distance by the sequence length. A lower Norm. DTW Distance indicates a higher degree of similarity between the generated audio and the given melody condition.

To further enhance our ablation studies, we have incorporated an additional objective metric: the Instrumentation Score (IS). This score is derived from multi-label instrument classification using F1 micro and F1 macro scores. The IS is crucial for assessing the models' capability to generate music that accurately represents the instruments specified in the textual descriptions. To obtain the IS, annotators, as described below, were tasked with listening to each sample generated by the model and annotating which instruments were audibly present in the loop.

**Subjective Evaluation Metrics.** Following the experimental setup in [17] and [3], we evaluate two subjective metrics: Overall Quality (OVL) and Relevance to Text Input (REL). Additionally, we introduce a new metric, Loop Consistency, to assess the seamlessness of the generated audio when played in a loop. Raters (annotators), recruited through Amazon Mechanical Turk, are required to have experience creating music with Digital Audio Workstations (DAWs) and experience in using or creating loops.

OVL assesses the perceptual quality of the audio samples, while REL measures how well the generated audio matches the given text input. Loop Consistency evaluates the smoothness of transitions when the audio is played in a continuous loop. All metrics are rated on a scale from 1 to 100, with higher scores indicating better performance.

We evaluate randomly sampled files, each assessed by a minimum of five raters to ensure reliable results. The CrowdMOS package is employed to identify and remove noisy annotations and outliers, following the guidelines provided in [25] and adopted by [3].

## 5 RESULTS

We present the evaluation results of our proposed Loops Transformer model in comparison with the state-of-the-art baseline, MusicGen [3]. Table 1 summarizes the performance of both models across various objective and subjective metrics. We consider three different settings for each model: (1) text-only conditioning, (2) text and full melody/MIDI conditioning, and (3) text and half length melody/MIDI conditioning.

In terms of objective metrics, the Loops Transformer achieves lower FAD scores, indicating that the generated audio is closer to real audio in terms of its distribution. When conditioned on both text and MIDI, the Loops Transformer obtains the lowest Normalized DTW Distance of 0.11, demonstrating its ability to generate audio that closely matches the given MIDI melody. Interestingly, the Loops Transformer achieves the best FAD score of 2.5 when conditioned on text and half length MIDI, suggesting that partial MIDI information can be sufficient for generating high-quality loops audio.

For the CLAP score, assessing the semantic alignment between generated audio and the corresponding text description, Music-Gen exhibits a marginal advantage over the Loops Transformer. This discrepancy can likely be ascribed to the training data each model was exposed to: while MusicGen was trained on a vast and diverse dataset encompassing a wide array of musical pieces, the Loops Transformer was specifically trained on loop-based data. This specialized focus on loops, although advantageous for generating cohesive and loopable segments, may not encompass the broader musical context to the same extent as MusicGen's training, influencing its performance on text-audio semantic alignment.

In the subjective assessments, the Loops Transformer markedly outperforms MusicGen in all dimensions. Notably, with conditions involving both text and MIDI, it achieves top marks in OVL (88.25), REL (85.96), and Loop Consistency (88.79). These results highlight the Loops Transformer's capacity for generating audibly superior loops that more accurately embody the specified textual descriptions and musical conditions and demonstrate improved loop coherence compared to MusicGen.

Interestingly, despite the Loops Transformer's lower performance on the CLAP score, indicative of text-audio semantic congruence, it achieves higher relevance ratings. This discrepancy may be attributed to textual prompts frequently specifying instruments (e.g., "A dynamic string loop set at 150 BPM in G minor..."), where accurate instrument representation may influence subjective relevance assessments. This contrasts with the CLAP metric potentially limited capacity to evaluate instrumentation accuracy, highlighting a discernible gap between algorithmic and human judgments regarding musical authenticity and textual congruity.

| Model | IS (f1 micro) | IS (f1 macro) |
|---|---|---|
| MusicGen | 55.61 | 53.28 |
| Loops Transformer w.o. IARL | 66.74 | 60.23 |
| Loops Transformer w. IARL | **74.39** | **71.26** |

Table 2: Evaluation of Instrumentation Score (IS) using f1 micro and f1 macro scores, derived from multi-label instrument classification. This table methodically compares MusicGen, Loops Transformer without Instrument-Aware Reinforcement Learning (IARL), and Loops Transformer with IARL, highlighting their performance in generating audio that precisely matches the instruments detailed in the input text.

| Model | $FAD_{vgg} \downarrow$ | CLAP $\uparrow$ |
|---|---|---|
| Loops Transformer (Text Only: T5) | 3.1 | **0.22** |
| Loops Transformer (MSCA w.o. T-to-M) | 2.8 | **0.22** |
| Loops Transformer (MSCA) | **2.6** | **0.22** |

Table 3: Performance comparison of Loops Transformer configurations. The configurations include the model with text input only using T5 embeddings (Text Only: T5), with Multi-Stage Cross Attention (MSCA), and MSCA without the integration of the Text-to-MIDI model (MSCA w.o. T-to-M).

| Model (1/8 data) | $FAD_{vgg} \downarrow$ | Norm. DTW Distance $\downarrow$ |
|---|---|---|
| w.o. loop shift | 3.3 | 0.17 |
| w. loop shift | **3.1** | **0.16** |

Table 4: Comparison of Loops Transformer performance with and without loop shift data augmentation on a reduced dataset (1/8 of the original pre-training dataset).

Table 2 presents the evaluation results for the Instrument-Aware Reinforcement Learning (IARL) approach. We compare the instrumentation scores (IS) of the Loops Transformer with and without IARL, as well as MusicGen. The Loops Transformer with IARL achieves the highest f1 micro and f1 macro scores, indicating its superior performance in generating audio with instruments that match the textual prompts. This highlights the effectiveness of the proposed IARL training strategy in enhancing the model's instrumentation capabilities.

To assess the impact of the Multi-Stage Cross Attention (MSCA) mechanism and the Text-to-MIDI Transformer, we evaluate different variations of the Loops Transformer using the FAD and CLAP metrics, as shown in Table 3. The Loops Transformer with MSCA achieves the best FAD score of 2.6, demonstrating the effectiveness of the proposed attention mechanism in integrating text and MIDI conditions. Removing the Text-to-MIDI Transformer component results in a degradation in performance, emphasizing its importance in the overall architecture.

Finally, Table 4 presents the performance of the Loops Transformer with and without the loop shift data augmentation technique on a reduced dataset (1/8 of the original pre-training data). The model trained with loop shift achieves better FAD and Normalized DTW Distance scores, highlighting the benefits of the proposed data augmentation strategy in improving the quality and consistency of the generated loops.

Overall, our experimental results demonstrate the superiority of the proposed Loops Transformer model in generating high-quality, relevant, and consistent music loops compared to the state-of-the-art baseline. The effectiveness of the key components, such as the Multi-Stage Cross Attention mechanism, Instrument-Aware Reinforcement Learning, and loop shift data augmentation, is validated through comprehensive ablation studies.

# 6 CONCLUSION AND FUTURE WORK

In this work, we introduced Loops Transformer, a novel controllable music loops generation model that integrates text and MIDI inputs to generate high-quality, relevant, and consistent music loops. We proposed a Multi-Stage Cross Attention mechanism to effectively combine textual and musical information, an improved Codebook Interleaving Pattern with Loop Shift data augmentation for seamless looping, and an Instrument-Aware Reinforcement Learning strategy to enhance the model's instrumentation capabilities.

Extensive experiments demonstrated the superiority of the Loops Transformer compared to the state-of-the-art baseline, MusicGen, across various objective and subjective metrics. The proposed model achieved significant improvements in generating music loops that closely match the given text and MIDI conditions, exhibit better perceptual quality, and maintain loop consistency. Ablation studies validated the effectiveness of the key components in contributing to the overall performance of the Loops Transformer.

Our work addresses the challenges in generating music loops that align with the detailed requirements of contemporary music production, offering a more interactive and adaptable creative framework for composers and music producers. The Loops Transformer empowers users to create customized loops by combining textual prompts with specific MIDI settings, accelerating the experimentation and exploration phases of music creation.

Future research directions include incorporating audio input as a conditioning signal, expanding the model's capability to generate longer musical sequences, and developing an interactive refinement process for music loops generation. The Loops Transformer represents a significant step towards bridging the gap between the creative vision of composers and the technical requirements of modern music production, offering a powerful and adaptable solution for generating high-quality, relevant, and consistent music loops.

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
