# OpenReview forum: "Controllable Music Loops Generation with MIDI and Text via Multi-Stage Cross Attention and Instrument-Aware Reinforcement Learning"
_acmmm.org/ACMMM/2024/Conference — MM2024 Poster_

### Official Review · Reviewer_GfhW · 2024-05-07

**Rating:** 4
**Confidence:** 3

**Summary:**

This paper proposes a Loops Transformer that can integrate text and MIDI inputs to generate music loops. Multi-Stage Cross Attention mechanism, Instrument-Aware Reinforcement Learning, and Codebook Interleaving Pattern with Loop Shift are used for effectively combining textual and musical information, enhancing the model’s instrumentation capabilities, and augmenting data, respectively.

**Strengths:**

1. The authors collected the MIDI-Audio Paired Loops Dataset to achieve the goal of this paper, i.e., generating loops given both textual descriptions and MIDI inputs. The MIDI input facilitates the precise rendering of critical music loop attributes.
2. A Multi-Stage Cross Attention mechanism is devised to integrate text and MIDI inputs.
3. The proposed Loops Transformer outperforms the state-of-the-art baseline model MusicGen in terms of loops generation.

**Limitations:**

1.	Missing references, the following studies are also relevant to loop generation:
* Symbolic Music Loop Generation with Neural Discrete Representations
* Context-aware Generation of Melodic MIDI Loops
* A Benchmarking Initiative for Audio-Domain Music Generation Using the Freesound Loop Dataset
2.	It's still not clear how $\mathrm{C_{txt}}$, $\mathrm{C_{MIDI}}$, and $\mathrm{C_{final}}$ are computed.
3.	Incomplete sentence:
* Left line 487-488: "This training strategy endows the Loops Transformer with greater flexibility, …"
4.	Lack of explanation for introducing reinforcement learning:
* Why not directly use instrument classification loss, i.e., binary cross-entropy loss, and instead introduce reinforcement learning and use the PPO loss? The reviewer is curious about the experimental results of using the instrument classification loss.
5.	Why do the subjective metrics include only Relevance to Text Input and not Relevance to MIDI/Audio Input?

**Suitability:**

3

---

### Official Review · Reviewer_v4HH · 2024-05-09

**Rating:** 4
**Confidence:** 3

**Summary:**

This paper introduces the Loops Transformer, a new framework for generating music loops by combining text and MIDI inputs. It employs a Multi-Stage Cross Attention mechanism to incorporate both textual and musical data, an enhanced Codebook Interleaving Pattern for seamless looping, and Instrument-Aware Reinforcement Learning to improve instrumentation in generated songs. Comparative experiments show that the Loops Transformer outperforms other state-of-the-art text-to-music generators by generating loops that align with input conditions, have higher perceptual quality, and maintain consistency. Ablation studies confirm the importance of its key components in achieving better performance.

**Strengths:**

* This work is well motivated - generating loops from texts is an understudied problem. This work can potentially serve as a useful supporting tool for expert composers.
* This paper is generally well-written, with some minor flaws & stylewise issues (see the limitations below) that sometimes may influence reading.
* The proposed method is intuitive and elegantly resolves existing problems.

**Limitations:**

* Lines 751-753, more details of the three settings should be added (e.g. what does "half-length" refer to?)
* For the ablation studies (Tables 2, 3, 4), it seems the performance was not measured on human ratings. Do $FAD_{vgg}$ and $CLAP$ suffice to assess the quality of generated audio?
* For Table 4, why 1/8 of the pretraining data were used (why not 1/2, 1/4, 1/16, 1/32). Does reducing data always lead to performance drop? Is it possible that the selected 1/8 is of lower quality than others and therefore influences the performance?
* In the provided listenable samples, personally I feel that Loops Transformer (LT) tends to use simpler instrumentations and effects (e.g. fewer instruments, more plain instruments) compared to MusicGen (MG). This is noticeable especially under energetic music, for example, Example 1 under Textual Description only, where the result from LT seems less spirited than MG due to the instrumentation. In addition, in Example 2 under Textual Description Only, I feel that there's been a sudden change in LT's generation (approximately at 12s) that seems to belie the description "flow gently" compared to MG's generation. Therefore, I feel that while admittedly LT can provide finer-grained controls (e.g. keys, BPMs), the style-wise control and faithfulness to conditions do not exhibit strong improvements from MG.
* Some minor nitpicks:
	* Line 230, has --> have
	* Section 3 seems to have some traits of AI writing, with many flowery wordings (e.g. marks the commencement, materializes, characterizes, advancing to, is representative of,)
	* In 4.2, are the three datasets (in bold) original? Any citations of the source of these datasets?
	* Using "interestingly" (Lines 777, 801, etc) does not sound formal and academic.

**Suitability:**

2

---

### Official Review · Reviewer_w2eK · 2024-05-23

**Rating:** 2
**Confidence:** 3

**Summary:**

This paper addresses the limitations of current text-to-music generation models in accurately rendering essential music loop attributes like melody, rhythms, and instrumentation. It introduces a new approach through the Loops Transformer and Multi-Stage Cross Attention mechanism that integrates textual and MIDI inputs effectively. An Instrument-Aware Reinforcement Learning technique is also developed to ensure accurate instrumentation. The model demonstrates superior performance over the state-of-the-art, improving both objective and subjective metrics, indicating its ability to produce musically coherent loops that meet the demands of modern music production.

**Strengths:**

1. Importance of Research Topic: The research enhances multimedia by integrating text and music generation, advancing the interaction between textual descriptions and musical compositions. This integration enriches user experience and opens new avenues for creative expression.

2. Rigorous Technical Approach: The paper uses advanced techniques like Loops Transformer, Multi-Stage Cross Attention, and Instrument-Aware Reinforcement Learning. These methods improve the precision and coherence of music generation, aligning closely with textual and MIDI inputs.

3. Potential Application Scenarios: The model has broad implications in music industry. It can revolutionize music creation.

**Limitations:**

Questions Related to Figures 1 and 2:
1.1: What are the input and output of the model (MIDI, audio, text, or a combination)?
1.2: What is the main technical contribution of the work?
1.3: How do the figures relate to Section 3, especially the mathematical notations in the content and the figures?
1.4: What is the relationship between Figure 1 and Figure 2?

Questions Related to Datasets:
2.1: How do GPT-4 or Claude analyze and generate textual descriptions from MIDI or MP3, particularly for descriptive and subjective content?
2.2: How was the MIDI-Audio pair dataset collected, and is it open-sourced?

Questions in Experiment:
3.1: What does "half melody/half MIDI" mean?
3.2: What is the specific meaning of REL, and how is the relevance of the output to the text measured, considering detailed and subjective descriptions?
3.3: How do the authors prove in the experiments that the proposed model precisely renders melody, rhythm, and instrumentation, addressing the limitations of other models?

Overall Questions:
4.1: What is the difference between repetition and loops in music, and why are works discussing repetition structure not mentioned? What are the differences and similarities between these works and this work?
4.2: What is the difference between your work and other works such as MusicGen? How does your model generate results that better fit text descriptions?
4.3: Why is an instrument-aware mechanism needed to identify instruments when instrument information can be directly specified and modified if required?
4.4: What is the motivation for taking MIDI as input but outputting audio?
4.5: What is the specific controllability of the model, and how does it allow users to control the output? What scope of control do users have over the result?
4.6: In the demo, why are the generated results similar to the input with explicit instrument specifications, but different without explicit specifications, especially in examples 2, 3, and 4?


Overall, it is an interesting work with many insightful ideas, but due to the paper's structure and logic, it is challenging to identify and validate the main contributions of the work.

**Suitability:**

3

---

### Official Review · Reviewer_zCiK · 2024-05-24

**Rating:** 5
**Confidence:** 2

**Summary:**

This paper presents a Transformer-based system for generating music that can be controlled via text and MIDI inputs. The authors introduce a Multi-Stage Cross Attention mechanism to improve the controllability of both types of input. Additionally, they employ an Instrument-Aware Reinforcement Learning technique to ensure accurate instrument arrangement. The results indicate that the proposed system can produce music loops that align closely with both text and MIDI inputs, maintaining correct instrumentation throughout.

**Strengths:**

1. The paper identifies a significant issue in current text-to-music generation systems: the lack of precise rendering of music loop features such as melody, rhythm, and instrumentation.
2. To address this, the authors propose a Multi-Stage Cross Attention mechanism and an Instrument-Aware Reinforcement Learning technique, which together enhance controllability and ensure accurate instrumentation.
3. The paper includes thorough and comprehensive experiments that demonstrate the effectiveness of the proposed controllable music generation system.

**Limitations:**

1. The quality of the figures in this paper is poor; please provide higher-resolution versions.
2. The equations should be centered on the line and labeled (e.g., (1), (2), (3), etc.). Additionally, if the equation is not at the end of a paragraph, a comma should be added at the end.
3. Since the dataset is self-generated, I would prefer to see more detailed statistical information. For instance, provide statistics on the text prompts (descriptions) created by ChatGPT-4 and Claude.
4. Although this is a minor issue, I'm curious about the rationale for using a 1-100 scale for MOS, as it can be challenging for respondents to distinguish between closely spaced scores (e.g., 75 vs. 76).
5. In the demo page you provided, I noticed that in the section "Textual Description and Musical Conditions (with Explicit Instrument Specification)," there is no demonstration for cases where the text description contains multiple instruments.

**Suitability:**

3

---

### Meta-Review · Area_Chair_H69L · 2024-07-02

**Recommendation:** Accept (Poster)
**Confidence:** 3

**Metareview:**

The paper identifies a significant issue in current text-to-music generation systems: the lack of precise rendering of music loop features such as melody, rhythm, and instrumentation. To address this issue, the paper proposes a Multi-Stage Cross Attention mechanism and an Instrument-Aware Reinforcement Learning technique, which enhance controllability and ensure accurate instrumentation. Experimental results show the effectiveness of the proposed controllable music generation system. Despite above strengths, reviewers have also identified several weaknesses which include: 1) missing references for loop generation: 2) ill-defined terms such as Half melody/half MIDI; 3) the difference between repetition and loops in music should be clarified.
In consideration of all review comments including the authors' rebuttal, this is a borderline paper for MM. However, its merits slightly outweigh its weaknesses. I recommend an accept and strongly encourage the authors to take reviewers' critiques into consideration when preparing for the final submission, if this paper is accepted.